# Investigating the Antioxidant Efficiency of Tea Flavonoid Derivatives: A Density Functional Theory Study

**DOI:** 10.3390/ijms26062587

**Published:** 2025-03-13

**Authors:** Yingmin Hou, Yuxi Wang, Xiaofei Tan, Yi Wang, Wenzhi Li, Xianzhen Li

**Affiliations:** 1School of Biological Engineering, Dalian Polytechnic University, Dalian 116034, China; yingminhou@163.com (Y.H.); txf20232024@163.com (X.T.); xianzhen@dlpu.edu.cn (X.L.); 2Institute of Frontier Chemistry, School of Chemistry and Chemical Engineering, Shandong University, Qingdao 266237, China; yx.wang@mail.sdu.edu.cn

**Keywords:** tea flavonoid, antioxidant activity, density functional theory

## Abstract

In this study, the antioxidant activity of 25 natural tea flavonoids was evaluated using Density Functional Theory (DFT), which identified four flavonoids with strong antioxidant activity: *kaempferol*, *fisetin*, *quercetin*, and *myricetin*. The analysis of electronic structures confirmed the positive effects of the number of -OH groups and intramolecular hydrogen bonds on the reactivity toward radicals. Electronic properties and potential energy curves (PECs) confirmed this view. Additionally, the solvation effect analysis indicated that a polar environment enhanced the antioxidant capacity of the products. Flavonoids exhibiting a Hydrogen Atom Transfer (HAT) mechanism, as the preferred antioxidant mechanism, show great potential for applications in fields such as biology, medicine, and food.

## 1. Introduction

The leaves of tea originating from the Yunnan–Guizhou Plateau in China are rich in flavonoids, which have an excellent antioxidant capacity and multiple physiological functions [1]. Flavonoids are usually combined with oxhydryl, methoxyl, and glucoside, showing significant anti-inflammatory, antibacterial, antioxidant, and anti-tumor activities [2,3,4,5]. So far, more than 4000 flavonoids have been isolated and identified [6,7]. As naturally efficient antioxidants, flavonoids can neutralize reactive oxygen species by transferring hydrogen or electrons to delay or prevent the oxidation of substances. Studies have confirmed that the molecular structure of flavonoids is closely related to the ability of scavenging free radicals [8,9,10].

While experimental studies have provided valuable insights into the antioxidant activity of flavonoids, their underlying molecular mechanisms remain poorly understood, due to the complexity of the interactions involved. Traditional experimental methods often face challenges in precisely identifying the reactive sites and understanding the role of molecular properties, such as electronic structure, hydrogen bonding, and solvation effects, in antioxidant reactions. Unlike experimental techniques, Density Functional Theory (DFT) allows us to analyze the interaction between flavonoids and free radicals visually, including the effects of -OH group positioning, intramolecular hydrogen bonds, and the electronic properties of the molecules [11]. Thus, computational assessment is the simplest method for the preliminary evaluation of natural antioxidants. 

In recent years, based on quantum biochemistry, the antioxidant properties and potential application prospects of flavonoids [12,13,14,15] have been comprehensively discussed. For phenolic compounds, the antioxidant mechanisms include Hydrogen Atom Transfer (HAT), Proton-Coupled Electron Transfer (PCET), Single-Electron Transfer followed by Proton Transfer (SET-PT), Sequential Proton Loss Electron Transfer (SPLET), Radical Adduct Formation (RAF), and Sequential Proton Loss Hydrogen Atom Transfer (SPLHAT) [16,17,18]. For the antioxidative compound Ar-OH, the enthalpies are defined as follows:(1)$$BDE=H\left(Ar-O\cdot \right)+H\left(H\cdot \right)-H(Ar-OH),$$(2)$$IP=H({Ar-OH}^{+}\cdot )+H(e^{-})-H(Ar-OH),$$(3)$$PDE=H\left(Ar-O\cdot \right)+H\left(H^{+}\right)-H({Ar-OH}^{+}\cdot ),$$(4)$$PA=H\left({Ar-O}^{-}\right)+H\left(H^{+}\right)-H(Ar-OH),$$(5)$$ETE=H\left(Ar-O\cdot \right)+H\left(e^{-}\right)-H({Ar-O}^{-}),$$

Computational evaluation mechanism analysis is beneficial for studying structure–activity relationship, and offers predictions for how molecular modifications could enhance antioxidant properties. Sagar Bag et al. summarized the flavonoids commonly found in tea and their biological activities, providing a new perspective on tea phytochemicals and their overall health benefits [19]. However, the mechanism of antioxidation and the relationship between antioxidant activity and structure have yet to be comprehensively studied. In this study, the bond dissociation enthalpy (BDE), ionization potential (IP), proton dissociation enthalpy (PDE), proton affinity (PA), and electron transfer enthalpy (ETE) parameters of 25 kinds of flavonoids in tea (Figure 1) were calculated by the DFT method. Their optimized neutral structure, ions, and free radical molecules were compared and analyzed to elucidate the relationship between structure and free radical scavenging activity. Electronic property analysis, frontier molecular orbitals (FMOs), and spin density distribution (SDD) were used to verify the above views. The free radical reaction of hydrogen peroxide (HOO·) was further evaluated to investigate the antioxidant properties. The study provides an essential computational perspective to screen natural strongly antioxidant products, filling the gap in theoretical data on the structure–activity relationships of tea flavonoids, and guiding future experimental designs for improving flavonoid-based antioxidants.

## 2. Results and Discussion

### 2.1. Geometrical Structures

The 25 different kinds of flavonoids were optimized in gas, water, and ethanol mediums at the B3LYP/Def-TZVP level (shown in Appendix A). Compared to the neutral molecules, the geometric structures of the phenoxy radicals did not undergo a significant change. A similar phenomenon was observed for Ar-O^−^ and Ar-OH^+^· produced by proton and electron extraction. According to the mechanism of the hydrogen abstraction reaction of polyphenol compounds, a planar structure usually facilitates the conjugation effect of the electrons within the molecule, thereby stabilizing the electron cloud and enhancing its radical reactivity [20]. Thus, flavonoids with larger planar structures, such as flavones (**1**–**6**) and flavonols (**7**–**12**), have a stronger electron donation ability, which helps to more effectively transfer hydrogen atoms or electrons and scavenge free radicals, meaning that they exhibit higher antioxidant activity. However, it is worth noting that the geometry of compound **12** shows a tendency of the B ring to be perpendicular to the A and C rings, which may reduce its antioxidant capacity. Furthermore, the presence of a methoxy group on the A ring, a 3-OH group on the B ring, a carbonyl group at the C4 position, a C2=C3 double bond, and intramolecular hydrogen bonds on the C ring all contribute to the higher stability and antioxidant capacity of these five conjugated tea flavonoid systems.

From a geometric perspective, the solvent polarity has a significant effect on the planar geometry and antioxidant capacity of the molecules. The polarity of the solvent affects the geometric structure of the molecule through solvation effects, especially for molecules with partially planar structures. For flavan-3-ols (**13**–**16**), flavanones (**17**–**21**), and flavanonols (**22**–**25**), as solvent polarity increases, the molecular conformation undergoes slight changes in solution, which leads to changes in the planarity and electron distribution. However, for fully planar compounds, the effect of solvent polarity on their geometric structure can be considered negligible.

### 2.2. Antioxidant Mechanisms

#### 2.2.1. HAT Mechanism

HAT is a fundamental step in the direct generation of free radical intermediates. In the free radical scavenging reaction, antioxidants directly donate H atoms to active free radicals, leading to the termination of radical chain reactions. In this process, BDE is commonly used to evaluate the ease of reactions [21]. At the B3LYP/TZVP level, all possible O-H bond BDE values of flavonoids were calculated (listed in Appendix A). Generally speaking, a lower BDE value is generally more conducive to free radical scavenging reactions. As shown in Table 1, the BDEs of compounds **7**–**11** and **19** were relatively low, ranging from 54.6 to 72.9 kcal/mol. The lowest BDE value was found at the C4′-OH bond of compound **8**. Compounds **7**, **9**, and **10** had the lowest BDE value at the C3-OH position. In addition, the O atom of the A ring should release a single electron to other groups forming the radical. The free radicals formed by dissociating C3-OH bonds can promote the antioxidant activity of flavonoids [22]. This is due to the intermolecular hydrogen bonds between the H atom of C3-OH and the O atom of C4=O, helping to promote the stability of free radicals. As shown in Appendix A, the optimized geometry for compounds **7**–**11** is similar between the neutral compounds and free radicals (dissociating the C3-OH), which shows that the C3-OH group can help to generate stable free radical molecules. These molecules will not be highly reactive due to a sharp structural change, and thus will not immediately trigger rapid oxidation reduction reactions. The C3-OH plays a key role here. 

It is well known that compound **10** (*Quercetin*), as a typical phenolic compound, is recognized to have strong antioxidant activity [23,24,25]. In the gas phase, for compounds **7**–**11** and **19**, the hydrogen supply capacity of the O-H bond followed the order **7** < **9** < **19** < **10** < **11** < **8**, and the BDE values of compounds **7**, **9**, **19**, and **10** were lower than 70 kcal/mol. Therefore, compounds **7** and **9**–**11** have higher antioxidant properties, and can be further studied as natural strongly antioxidant substances.

In order to study the effects of different dielectric environments on the hydrogen supply capacity, the BDEs of 25 compounds in the ethanol and water solvents were calculated (listed in Appendix A). In ethanol solvent, the structures of the radicals with the lowest BDEs for Flavan-3-ol derivatives (**13**–**16**) and dihydroflavonol derivatives (**22**–**25**) differed from those in the gas phase. In contrast, the structures of the radicals with the lowest BDEs for flavonol derivatives (**7**–**12**) were the same as those in the gas phase. In an aqueous environment, the structures of the radicals with the lowest BDEs for all 25 compounds were different from those in the gas phase. It can be seen that the dielectric environment affected the antioxidant activity of the flavonoid radicals. Generally, as the solvent polarity increased, the BDEs also increased, with the increase remaining almost within the range of 24.4 ~ 38.1 kcal/mol. The increased BDEs suggest that the HAT process became more difficult, weakening the ability of flavonoids to donate hydrogen atoms to free radicals, thereby reducing their antioxidant activity.

#### 2.2.2. SET-PT Mechanism

For the SET-PT mechanism, the calculated IPs and PDEs are the two most critical thermodynamic parameters (Table 2). The higher the electron delocalization of flavonoids, the stronger the electron donor ability is. In the gas phase, flavonols **7**–**11** had the lowest (IP+PDE) min values at the 3-OH and 4′-OH positions, which proves that flavonols have a strong electron-donating ability. Compounds **8** and **11** had the lowest IP values, of 125.1 kcal/mol and 124.8 kcal/mol, respectively. Generally, the SET-PT mechanism mainly determines the transfer of electrons from Ar-OH [26]. The decrease in ionization energy may be due to the formation of intramolecular hydrogen bonds between C5-OH and C4=O. In addition, it is related to the electron delocalization caused by the formation of the double bond of C2=C3 in the B ring. According to the data in Table 2, under the same conditions, the IP value of dihydroflavonol decreases with the increase in the number of OH groups (**24** < **23** < **25** < **22**). Therefore, the strong antioxidant activity may be related to the number of phenolic hydroxyl groups in the B ring.

In order to further assess the impact of the dielectric environment on oxidation resistance, the IP and PDE values in the ethanol and water phases were calculated at the same computational level (Table 2 and Appendix A). In contrast to the BDEs, the IPs of most compounds, except for compounds **8** and **10**–**12**, slightly decreased under polar conditions. The PDEs increased in the range of 29.6 ~ 38.2 kcal/mol. Notably, the IP and PDE values in the gas phase, ethanol medium, and water phase were all higher than their respective BDE values, with the combined IP + PDE energy approximately 3.6 times that of the BDEs. Compared to the SET-PT mechanism, the HAT mechanism appears to be the primary pathway for flavonoids in free radical scavenging.

#### 2.2.3. SPLET Mechanism

PAs and ETEs are the main thermodynamic parameters used to evaluate the difficulty of the SPLET mechanism. Consistently with the conclusion based on the HAT mechanism, the (PA+ETE) _min_ of compounds **7**, **9**, **10** (3-OH), and **11** (4′-OH) in the gas phase was lower than the minimum values of the other compounds (Table 3), which were 383.9 kcal/mol, 384.4 kcal/mol, 382.0 kcal/mol, and 377.4 kcal/mol, respectively. The results show that these compounds have strong antioxidant activity. In addition, as the number of O-H groups in the B ring increased, the PAs decreased. This means the number of O-H groups in B ring also significantly affects the ionization energy of flavonoid compounds [27].

The first step in the SPLET pathway is highly sensitive to environmental changes. Consequently, the PAs and ETEs of the studied compounds in ethanol and water were calculated (Table 3, Appendix A). Compared to the gas phase, the PAs decreased by less than 49.6 kcal/mol, and the ETEs increased slightly, by less than 45.8 kcal/mol, as the solvent polarity increased, with the exception of compound **11**, which showed an increase. These results suggest that electron loss is more favorable in a strong dielectric environment than in the gas phase, and that the deprotonation process is more efficient than the single electron transfer process.

In addition, PA+ETE values had a relationship with BDE of the same magnitude as that of IP+PDE in the gas phase, ethanol solvent, and water phase, which indicates the SPLET mechanism was not favored in any of the research environments. The HAT mechanism seems to be the main way in which flavonoids scavenge free radicals. Based on the above analysis, compounds **7**, **9**, and **10** were further investigated to examine the antioxidant capacity of flavonoids.

### 2.3. Frontier Molecular Orbital Theory and Spin Density Distribution

Hydrogen, electrons, or a combination of both are involved in all of the studied antioxidant mechanisms. In the process of radical scavenging, the transfer of the phenolic hydroxyl hydrogen of flavonoid molecules is accompanied by electron transfer [13]. The highest occupied molecular orbital (HOMO) and the lowest unoccupied molecular orbital (LUMO) energies of antioxidants are key quantum descriptors for electron transfer in antioxidants, playing a crucial role in the chemical reactivity of antioxidants [28]. Huai Cao et al. pointed out that a lower HOMO energy indicates a weaker electron-donating ability at the scavenging site [29]. In contrast, a higher HOMO energy suggests that the scavenging site is a promising electron donor, with strong free radical scavenging efficiency. As shown in Figure 1, in the gas phase, the HOMO energy (E_HOMO_)for compound **11** (−6.080 eV) was lower than that of compounds **7** (−6.046 eV), **9** (−6.036 eV), and **10** (−5.952 eV). The results show that compound **11** had the strongest antioxidant capacity among the studied products. It is worth noting that polar solvents have less of an effect on E_HOMO_, but reduce the LUMO energy (E_LUMO_). 

The FMO theory indicates that the transition state in chemical reactions forms due to the interaction between the HOMO and LUMO orbitals of antioxidants and free radicals. The HOMO–LUMO gap (E_LUMO-HOMO_) of compounds helps to predict their antioxidant strength and stability [30]. The E_(LUMO-HOMO)_ of the compound in the gas phase follows **10** > **9** > **7** > **11**. The results show that increasing the number of -OH groups is beneficial for decreasing E_(LUMO-HOMO)_. Although compound **7** and compound **9** had the same number of -OH, the E_(LUMO-HOMO)_ was different. This is due to the formation of intramolecular hydrogen bonds between the carbonyl groups at C5-OH and C4, which is beneficial for enhancing antioxidant activity. The results of the ethanol and aqueous phase provide a reasonable basis for the reduction in the E_(LUMO-HOMO)_ values of the compounds. 

Under polar conditions (Appendix A), the E_(LUMO-HOMO)_ of compounds **7** and **9**–**11** was reduced, which is beneficial to the removal of free radicals. Under polar conditions (Appendix A), the E_LUMO-HOMO_ values of compounds **7** and **9**–**11** decreased, suggesting that polar environments favor the scavenging of free radicals. Except for compound **7**, the E_HOMO_ and E_LUMO_ energies of the other compounds decreased, leading to a reduction in E_LUMO-HOMO_. The position of the hydroxyl substituents and the formation of intramolecular hydrogen bonds play a key role in determining the strength of the solvation effect. Compared to compound **7**, the hydroxyl group on the B ring of compound **9** had a stronger influence on antioxidant capacity than those on the A and C rings. The presence of multiple hydroxyl groups increases the number of intramolecular hydrogen bonds, which, in turn, affects the interaction between the solvent and the molecule.

In addition, to further explore the details of the solvation effect and reduce computational costs, we considered the complexes of compounds **7** and **9**–**11** with water and ethanol, respectively (Appendix A), and discussed them using an implicit model. The ordering of the E_LUMO-HOMO_ gaps for the complexes was consistent with that of the monomers, i.e., the antioxidant capacity of compounds was in the order of **10** > **9** > **11** > **7**. Increasing the solvent polarity enhanced the antioxidant capacity of the complexes of compounds **7** and **9**–**11**. Our study clearly demonstrates that the interactions between solute and solvent molecules, as modeled by the implicit model chosen in this study, can accurately identify natural products with strong antioxidant potential.

SDD provides information about the rate of free radical scavenging reactions. Notably, the higher the delocalized SDD, the easier the formation of free radicals, indicating that the compound exhibits a faster free radical scavenging reaction rate [31]. Figure 2 shows the calculation of the atomic SDD of each radical after hydrogen extraction of compounds **7** and **9**–**11** in the gas phase. Taking compound **7** as an example, the calculated results show that the order of atomic spin density values is 0.418 (7-OH) > 0.348 (4′-OH) > 0.347 (5-OH) > 0.315 (3-OH). The oxygen atom at the C3-OH site has a strong spin distribution, and the C3-OH site is a good attack site for the radical reaction of compound **7**. This conclusion is also well verified by the results for compounds **9**–**11**. As concluded above, the free radicals formed at C3 sites have strong antioxidant activity. In addition, in polar solvents, as the atomic spin density decreases (Appendix A), and corresponding to an increase in BDE values, the antioxidant activity decreases.

### 2.4. Electronic Properties

The electronic properties of molecules, such as chemical hardness (η), are closely linked to their antioxidant strength. η reflects a molecule’s resistance to external perturbations; compounds with higher hardness are less prone to attack, thus demonstrating stronger antioxidant properties. All the data of electronic properties are listed in Table 4. Compound **10** was much more stable than the others (η = 2.115 eV), especially compared with compound **11** (η = 1.997), which was more active. The η of compounds was lower in polarity than in the gas phase. These results again show that an increase in the number of phenolic hydroxyl groups can enhance the antioxidant strength of compounds.

The chemical potential (μ) and mulliken electronegativity (χ) represent the degree to which an atom attracts a pair of lone electrons. According to Sanderson’s principle, compounds with high electronegativity may quickly reach equilibrium and establish low reactivity [32]. In addition, χ indicates a molecule’s affinity for electrons. Molecules with higher electronegativity more readily acquire electrons, allowing them to efficiently scavenge free radicals and, thus, enhance their antioxidant efficacy. As shown in Table 4, the χ value of compound **11** is 4.083 eV in the gas phase, and its antioxidant activity is higher than that of compounds **7** (4.029 eV), **9** (3.975 eV), and **10** (3.818 eV). It can be seen that compounds with lower η and χ values are always related to better antioxidant activity. Compared with η, the value of χ in the non-polar gas phase is lower than that in the ethanol and water media. The overall electrophilic index (ω) reveals the ability of molecules to accept surrounding electrons [33]. When the solvent environment changes from gas to liquid, significant changes occur in ω, $\omega^{+}$, and $\omega^{-}$. In particular, in all media, ω is always twice as large as $\omega^{-}$, and $\omega^{+}$ is always three times as large as $\omega^{-}$.Therefore, flavonols themselves have strong electron donor properties.

Calculating the dipole moment is an effective method to estimate the separation of positive and negative charges in a molecule. High-order dipole moments are accompanied by high charge density and high-polarity bonds [34]. An increase in the dipole moment reduces the antioxidant efficiency of a compound, which is not conducive to the free radical scavenging reaction [35]. The antioxidant activity of the compounds followed the order of **11** > **9** > **7** > **10**, according to increasing antioxidant activity. In contrast, compounds **7** and **9**–**11** were 1.5 times as strong in the water and ethanol media as in the non-polar gas phase. The results show the polar solvents enhance the antioxidant capacity of flavonols.

### 2.5. Potential Energy Surface of HOO· Radical Scavenging

Although hydroxyl radicals (HO·) are highly reactive and can serve as experimental subjects for in-depth exploration of reaction mechanisms, their reaction rates are extremely fast, and the reactions are usually completed within the time scale of nanoseconds (ns) to microseconds (ms), which is not conducive to experimental observation. In contrast, the reaction rate of HOO· radicals is relatively slow, making it easier to control the experimental conditions. Moreover, their reaction mechanism is relatively simple, which is more beneficial for structural design and performance regulation.

To demonstrate the HAT mechanism as a potential primary antioxidant reaction mechanism in the gas phase, the reactions of compounds **7** and **9**–**11**, which had low BDEs, with HOO**·** free radicals was investigated. According to the transition state theory, determining the transition state and calculating the reaction rate from the height of the barrier help to provide an understanding of the reaction mechanism. Generally, the decomposition reaction is dominant, and can occur at room temperature if the barrier is less than 21 kcal/mol. In addition, via analyzing the activation energy of flavonoid compounds in antioxidant reactions, this method is helpful for evaluating their efficiency in scavenging free radicals. A lower energy barrier indicates a more effective antioxidant, as it suggests that the compound can readily undergo reactions to neutralize harmful free radicals. The energy barrier (Figure 3) of the compound decomposition reaction was in the order of compound **11** (4.3 kcal/mol) > compound **10** (1.6 kcal/mol) > compound **9** (1.5 kcal/mol) = compound **7**. This indicates that, for compounds **7**, **9**, and **10**, the hydrogen at the site with the lowest BDE value was more easily transferred to the HOO**·** radical to form the free radical (Ar-O**·**) and H_2_O_2_. This is consistent with the conclusions of the above analysis.

## 3. Materials and Methods

In this paper, the Gaussian 16 [36] package was used for all the calculations. Using the DFT method, we optimized the geometric structure and analyzed the frequency of 25 tea flavonoids, including their free radicals, free radical cations, and anions, at the B3LYP [37,38]/def-TZVP [39,40] level. This method has been successfully applied to studying the geometric structures of flavonoids [20,41,42,43]. The single-point energies of the compound molecules were also calculated at the same level. The D3 version of Grimme’s dispersion [44] was introduced to account for dispersion forces. This has been demonstrated to be necessary in previous benchmark studies on weak interactions [45].

The integral equation formalism (IEF) variant [38,46] of the polarizable continuum model (PCM) [47,48] was used to calculate the effects of ethanol (ε = 24.85) and water (ε = 78.35) on the properties of the compounds. Considering the details of the solvation effect, such as the specific interactions between solvent molecules and solute molecules, explicit solvent molecules can provide a more accurate solvation model. However, the computational cost of this method is relatively high, especially for large systems, making it potentially impractical. Therefore, in our study, a small number of solvent molecules were added around certain solute molecules (**7**, **9**–**11**), and an implicit solvation model was used to treat the system, in order to further investigate the impact of solvation effects on the antioxidant mechanism.

At the same level, the profiles evaluated included FMO analysis, spin density, η, μ, χ, and ω [49,50,51]. These chemical indicators are written as follows:(6)$$\eta \approx \frac{E_{LUMO}-E_{HOMO}}{2},$$(7)$$\mu \approx -\frac{E_{LUMO}+E_{HOMO}}{2}=-\chi ,$$(8)$$\omega =\frac{\mu^{2}}{2\eta },$$(9)$$\omega^{-}=\frac{{\left(3IP_{0}+EA\right)}^{2}}{\left[16\left(IP_{0}-EA\right)\right]},$$(10)$$\omega^{+}=\frac{{\left(IP_{0}+3EA\right)}^{2}}{\left[16\left(IP_{0}-EA\right)\right]},$$

In addition, the enthalpies of the hydrogen atom (H·), proton (H^+^), and electron (e^−^) also figure in these equations, where the enthalpies of the proton and electron cannot be calculated. For the gas phase, the thermodynamic quantities can be easily obtained by using the commonly accepted values of 6.1398 kJ/mol for the proton enthalpy and 3.1351 kJ/mol for the electron enthalpy [49,52].

## 4. Conclusions

Screening the antioxidant activity of natural products is critical for discovering new therapeutic agents, promoting sustainable solutions, ensuring safety, and meeting the increasing consumer demand for natural alternatives in healthcare, food, and cosmetics. In this study, the antioxidant activity of 25 natural tea flavonoids was evaluated using DFT methods, including the effects of polarity and geometric configuration on the HAT, SET-PT, and SPLET mechanisms. The study also produced the following findings:

(1) *Kaempferol*, *fisetin*, *quercetin*, and *myricetin* were identified as potential antioxidants.

(2) The planarity of the geometric structure helps to enhance the electron conjugation effect of the molecule, thereby increasing its antioxidant activity. Solvent polarity adjusts the geometry of the molecule through the solvation effect, thus affecting its antioxidant capacity.

(3) The antioxidant activity is positively correlated with the number of -OH groups on the B ring and the number of intramolecular hydrogen bonds in the compound.

(4) Intramolecular hydrogen bonding may be more significant, as flavonoids with two or three hydroxyl groups on the B ring preferentially transfer hydrogen atoms from the B ring rather than the A, C rings.

(5) The HAT mechanism is the primary pathway for the gas phase reactions of natural flavonoids. Compared to the SPLET and SET-PT mechanisms, the HAT mechanism is less affected by solvent polarity.

This research quantifies the antioxidant mechanisms of flavonoid compounds, providing a theoretical basis for understanding their free radical scavenging ability. This opens new avenues for the development of more efficient natural antioxidants.

## Data Availability

The authors confirm that the data supporting the findings of this study are available within the article [and/or] its Appendix A.

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
