# Peer review of "Investigating the Antioxidant Efficiency of Tea Flavonoid Derivatives: A Density Functional Theory Study"

_ijms, 2025, doi:10.3390/ijms26062587_

Round 1
Reviewer 1 Report
Comments and Suggestions for Authors
In this work, the authors performed used DFT calculations to assess the antioxidant activity of 25 natural flavonoids, focusing on their structure-activity relationship in both polar and gas phases. Four flavonoids—kaempferol, fisetin, quercetin, and myricetin—showed strong antioxidant activity. The research highlighted the positive influence of -OH groups and intramolecular hydrogen bonds on free radical reactivity, supported by frontier molecular orbitals (FMOs) and spin density distribution (SDD). Electronic properties and potential energy curves (PECs) further confirmed these findings. The study also considered solvation effects to understand how polar environments impact antioxidation mechanisms. Flavonoids with a hydrogen atom transfer (HAT) mechanism were identified as having significant potential for applications in biology, medicine, and food industries.
This work can be interesting for the computational chemistry and the industrial community. I would like to ask the authors to consider the minor comments below.
1. The long-range interaction can play an important role in the studied systems. Can the authors discuss the effects of not adding the dispersion correction in the DFT calculations?
2. In this work, did the authors combine implicit solvent model with explicitly add solvent molecules to describe the solvation effect?
3. Table 4
IP/EA and HOMO/LUMO give same information, redundant columns should be removed.
4. Figure 1
It should be mentioned in the context that orbital energies computed at the DFT level have errors and the starting point dependence.
5. The reference for the def2 basis set is missing: Phys. Chem. Chem. Phys. 7, 3297 (2005)
Author Response
Comments1: In this work, the authors performed used DFT calculations to assess the antioxidant activity of 25 natural flavonoids, focusing on their structure-activity relationship in both polar and gas phases. Four flavonoids—kaempferol, fisetin, quercetin, and myricetin—showed strong antioxidant activity. The research highlighted the positive influence of -OH groups and intramolecular hydrogen bonds on free radical reactivity, supported by frontier molecular orbitals (FMOs) and spin density distribution (SDD). Electronic properties and potential energy curves (PECs) further confirmed these findings. The study also considered solvation effects to understand how polar environments impact antioxidation mechanisms. Flavonoids with a hydrogen atom transfer (HAT) mechanism were identified as having significant potential for applications in biology, medicine, and food industries.
Response1: We thank the reviewer for the positive reception of our work.
Comments2: The long-range interaction can play an important role in the studied systems. Can the authors discuss the effects of not adding the dispersion correction in the DFT calculations?
Response2: We appreciate the valuable comments provided by the reviewer. Since dispersion forces are long-range interactions, they can be more accurately accounted for in DFT calculations using various dispersion correction methods (e.g., DFT-D3, DFT-D2, etc.). We acknowledge that the omission of these methods may lead to an underestimation of intermolecular interactions in certain molecules, especially in large, structurally complex natural flavonoid molecules. Therefore, the computational results without dispersion corrections might overlook some subtle effects, particularly the details of weak intermolecular interactions. To enhance the accuracy and reliability of our conclusions, we have reiterated this issue in the revised manuscript (in the Computational details section) and referenced relevant studies.
Comments3: In this work, did the authors combine implicit solvent model with explicitly add solvent molecules to describe the solvation effect?
Response3: Thank you for your careful review of our manuscript. In our study, we only used an implicit solvation model (e.g., the PCM model) to account for solvation effects, without explicitly adding solvent molecules. However, when considering the details of the solvation effect, such as the specific interactions between solvent molecules and solute molecules, explicit solvent molecules can provide a more accurate solvation model. Nevertheless, this comes with higher computational costs and may be impractical, especially for large systems. For this reason, we did not use explicit solvent molecules on a large scale in our manuscript.To address this issue, we added a small number of solvent molecules around four solute molecules (7, 9-11) with good antioxidant activity and then used the implicit solvation model to treat the system. The obtained data were analyzed and discussed, including the stability of the radical and the HAT mechanism, to further verify the impact of solvation effects on the antioxidant mechanism (Table 1). The energy gap ordering obtained under the two computational treatment methods is consistent. Research shows that the interactions between solutes and the solution do not affect the ranking of the overall antioxidant capacities. The results calculated under the implicit solvent model are reliable.
The relevant discussions are added to the manuscript and highlighted.
Table 1. Comparing calculated EHOMO, ELUMO, ELUMO-HOMO (in eV) datas for compound 7, 9-11 basis for different solvent methods at the B3LYP/TZVP level.
|
|
Implicit solvent |
EHOMO |
ELUMO |
ELUMO-HOMO |
|
7 Kaempferol |
GAS |
-6.046 |
-2.001 |
4.045 |
|
|
Water/No addition |
-6.052 |
-2.067 |
3.985 |
|
|
Water addition |
-6.263 |
-2.308 |
3.955 |
|
|
Ethanol/No addition |
-6.049 |
-2.060 |
3.989 |
|
|
Ethanol addition |
-6.258 |
-2.306 |
3.952 |
|
9 Fisetin |
GAS |
-6.036 |
-1.911 |
4.123 |
|
|
Water/No addition |
-6.041 |
-2.027 |
4.014 |
|
|
Water addition |
-6.243 |
-2.268 |
3.975 |
|
|
Ethanol/No addition |
-6.037 |
-2.017 |
4.020 |
|
|
Ethanol addition |
-6.247 |
-2.254 |
3.993 |
|
10 Quercetin |
GAS |
-5.952 |
-1.723 |
4.229 |
|
|
Water/No addition |
-5.952 |
-1.858 |
4.094 |
|
|
Water addition |
-6.189 |
-2.116 |
4.073 |
|
|
Ethanol/No addition |
-5.948 |
-1.846 |
4.102 |
|
|
Ethanol addition |
-6.090 |
-2.087 |
4.003 |
|
11 Myricetin |
GAS |
-6.080 |
-2.087 |
3.993 |
|
|
Water/No addition |
-6.133 |
-2.144 |
3.989 |
|
|
Water addition |
-6.365 |
-2.391 |
3.974 |
|
|
Ethanol/No addition |
-6.127 |
-2.136 |
3.991 |
|
|
Ethanol addition |
-6.343 |
-2.375 |
3.968 |
Comments 4:Table 4 IP/EA and HOMO/LUMO give same information, redundant columns should be removed.
Response 4:Thank you for your careful review of our manuscript. According to your suggestion, we modify form Table 4 to delete the duplicate information.
Comments 5: Figure 1 It should be mentioned in the context that orbital energies computed at the DFT level have errors and the starting point dependence.
Response 5: Thank you for the reviewer’s attention to this important issue. We completely agree that orbital energies in DFT calculations indeed contain some errors, and the results are affected by the choice of functional and basis set. Additionally, orbital energies in DFT calculations may also be influenced by starting point dependence, particularly when dealing with systems with strong electronic correlation. In our study, based on previous benchmark studies[1,2], we selected appropriate functionals and basis sets for the calculations to improve the accuracy of the results as much as possible. The B3LYP functional (HF=20%) combined with D3 dispersion corrections is able to effectively study the systems investigated (where all atoms are main-group elements), and yield the correct global minimum. The basis set determines the approximate form of the electronic wave function. Smaller basis sets may not accurately describe the molecular electronic structure, leading to calculation errors. Furthermore, the radical molecules studied in Figure 1 are all neutral molecules, and the effect of dispersion is almost negligible. To further validate this point, we also discuss the frontier molecular orbital results under the 6-311G(d,p) basis set in the revised manuscript. As shown in the Table 2, the dispersion effect has little effect on the results for flavonoid compounds. This finding is generally applicable to the selected research group.
Table 2. Comparing calculated EHOMO, ELUMO, ELUMO-HOMO (in eV) datas for compound 7, 9-11 basis on TZVP/6-311G(d,p) basis set.
|
|
|
EHOMO |
ELUMO |
ELUMO-HOMO |
|
7 Kaempferol |
TZVP |
-6.046 |
-2.001 |
4.045 |
|
|
6-311+G(d,p) |
-6.094 |
-2.079 |
4.015 |
|
9 Fisetin |
TZVP |
-6.036 |
-1.911 |
4.123 |
|
|
6-311+G(d,p) |
-6.087 |
-1.994 |
4.093 |
|
10 Quercetin |
TZVP |
-5.952 |
-1.723 |
4.229 |
|
|
6-311+G(d,p) |
-5.987 |
-1.789 |
4.198 |
|
11 Myricetin |
TZVP |
-6.080 |
-2.087 |
3.993 |
|
|
6-311+G(d,p) |
-6.135 |
-2.162 |
3.973 |
In addition, the starting point dependence issue is also of interest to us. In the study, we first performed preprocessing using a conformational search, and the obtained geometries were further optimized in Gaussian 16 to minimize errors. In future work, we will further discuss this issue.
Comments 6: The reference for the def2 basis set is missing: Phys. Chem. Chem. Phys. 7, 3297 (2005)
Response 6: Thank you for your careful review of our manuscript. According to your suggestions, we have examined the manuscript and corrected the references.

Reviewer 2 Report
Comments and Suggestions for Authors
Investigating the Antioxidant Efficiency of Tea Flavonoid Derivatives: A DFT Study
This article presents a Density Functional Theory (DFT) study on the antioxidant efficiency of 25 natural tea flavonoids. There are several areas that require improvement to enhance the clarity, accuracy, and impact of the manuscript.
Comments:
1) The title of the paper needs to be rewritten and abbreviations avoided.
2) Abstract:
- The abstract should be more concise and directly mention the specific flavonoids with strong antioxidant activity.
3) Introduction:
- The research gap needs to be clearly stated. Please explain why a DFT study is necessary and how it complements existing research.
4) Computational Details:
- Provide more information on the validation of the computational methods used. Mention if these methods have been validated for similar compounds.
5) Results and discussion
- Results not well presented and discussed. This section needs further deliberation (cite relevant contextualizing references). Rationalization for the observed results was not given due attention. Please revise the relevance of the findings in the light of other comparable studies.
6) Tables and Figures:
- Simplify the tables by summarizing key data in the text and moving full tables to supplementary materials. Improve figure labels and units for clarity.
7) Antioxidant Mechanisms:
- Elaborate on how the solvent environment affects the antioxidant mechanisms. Provide specific details on solvent-flavonoid interactions.
8) Frontier Molecular Orbital Theory and Spin Density Distribution:
- Clarify the interpretation of FMOs and SDD results. Explain the significance of HOMO and LUMO energy levels in antioxidant activity.
9) Electronic Properties:
- Explain how electronic properties like chemical hardness and electronegativity relate directly to antioxidant strength.
10) Potential Energy Surface of HOO· Radical Scavenging:
- Discuss the practical implications of the energy barriers for using these flavonoids as antioxidants.
11) Conclusion:
- Highlight potential applications and suggest future research directions. Mention the possibility of synthesizing derivatives to enhance antioxidant properties.
Comments on the Quality of English Language
The manuscript requires significant editing to improve the clarity, grammar, and overall quality of the English language.
Author Response
Comments 1: The title of the paper needs to be rewritten and abbreviations avoided.
Response 1: Thanks for the kind suggestion. We made adjustments and revised them in the manuscript.
Comments2: Abstract:- The abstract should be more concise and directly mention the specific flavonoids with strong antioxidant activity.
Response 2: Thanks for the reviewer’s careful reading. In the revised abstract, we more directly mentioned the specific flavonoids with strong antioxidant activity, such as kaempferol, fisetin, quercetin, and myricetin, and simplified certain contents to enhance the conciseness and informativeness of the abstract.
Comments 3: Introduction:- The research gap needs to be clearly stated. Please explain why a DFT study is necessary and how it complements existing research.
Response 3: Thanks for the kind advice. According to your suggestions, we have revised the introduction section. Specifically, the following changes were made: the current state of research was added, the advantages of DFT studies were reiterated, and the necessity of theoretical methods in the study of flavonoid compounds was emphasized. The relevant descriptions have been revised and highlighted in the manuscript.
Comments 4: Computational Details:- Provide more information on the validation of the computational methods used. Mention if these methods have been validated for similar compounds.
Response 4: Thank you for the reviewer’s valuable comments on the validation of the computational methods. The Density Functional Theory (DFT) method we employed, particularly the B3LYP exchange-correlation functional and def-TZVP basis set, has been widely applied in the study of similar natural products and flavonoid compounds, with numerous publications demonstrating its effectiveness and accuracy for these types of compounds. We will provide a detailed explanation of the application systems and advantages of these methods, citing relevant literature to support our choice of methodology.
Comments 5: Results and discussion- Results not well presented and discussed. This section needs further deliberation (cite relevant contextualizing references). Rationalization for the observed results was not given due attention. Please revise the relevance of the findings in the light of other comparable studies.
Response 5: Thank you for the reviewer’s valuable comments on the “Results and Discussion” section. We acknowledge that rationalizing our observed results and comparing them with related studies is crucial for enhancing the quality of this section. Therefore, we will supplement and improve the "Results and Discussion" section in the revised manuscript, adding more background discussion and references to better explain and justify our observed results. Specifically, in this section, we will:
(1) Supplement relevant literature: We will cite more related research, compare our results with previous studies, and clarify the unique contribution of our research.
(2) Discuss results in more depth: For the observed antioxidant activities, particularly the performance of quercetin, fisetin, kaempferol, and myricetin, we will provide a more detailed rationale, explaining why these compounds exhibit strong antioxidant activity in this study.
(3) Rationalize and compare with literature: We will more clearly articulate the correlations we discovered and compare them with findings from other comparable studies, discussing the significance and practical implications of our results. For example, how similar DFT studies and experimental research support the trends we observed, or the novelty of our findings.
Comments 6: Tables and Figures:- Simplify the tables by summarizing key data in the text and moving full tables to supplementary materials. Improve figure labels and units for clarity.
Response 6:Thanks for the kind suggestion. We agree that simplifying the tables, improving the clarity of the figures, and moving the detailed data to the supplementary materials are important measures to enhance the quality of the paper. In the revised manuscript, we will make the following changes according to the reviewer’s suggestions: we will present only the key information from the tables in the main text and move the more detailed data to the supplementary materials, so that readers who need more detailed information can refer to them. This will make the main text more concise and clearer.
Comments 7: Antioxidant Mechanisms:- Elaborate on how the solvent environment affects the antioxidant mechanisms. Provide specific details on solvent-flavonoid interactions
Response 7:Thanks for the reviewer's suggestions. The solvent environment plays a crucial role in the antioxidant reaction, especially the interaction between solvent molecules and flavonoid molecules. In the manuscript, we further discuss the effects of solvation on electronic structure, geometry, etc., based on current implicit solvent studies, and then analyze the antioxidant reaction pathways.
Comments 8: Frontier Molecular Orbital Theory and Spin Density Distribution:- Clarify the interpretation of FMOs and SDD results. Explain the significance of HOMO and LUMO energy levels in antioxidant activity.
Response 8: Thank the reviewers for their suggestions. We will provide a more detailed explanation of the FMOs and SDD results in a revised draft, and explain the significance of HOMO and LUMO levels in antioxidant activity. At the same time, we also cite relevant studies to demonstrate.
Comments 9: Electronic Properties:- Explain how electronic properties like chemical hardness and electronegativity relate directly to antioxidant strength.
Response 9:Thanks for the reviewer's suggestions. Chemical hardness and electronegativity are very important in explaining the role of antioxidant activity, especially in terms of molecular reactivity and stability. In the revised draft, we will explain in detail how these electronic properties affect the antioxidant strength of flavonoid molecules.
Comments 10: Potential Energy Surface of HOO· Radical Scavenging:- Discuss the practical implications of the energy barriers for using these flavonoids as antioxidants.
Response 10: Thank the reviewers for the valuable comments. In the revised draft, we will discuss the energy potential surface analysis results in detail and illustrate how energy barriers affect the practical use of flavonoids as antioxidants, including discussing the relationship between energy barriers and reaction rates, emphasizing the relationship between energy barriers and antioxidant applications.
Comments 11: Conclusion:- Highlight potential applications and suggest future research directions. Mention the possibility of synthesizing derivatives to enhance antioxidant properties.
Response 11: Thanks for the kind suggestion. In the revised conclusion, we will emphasize the potential of flavonoid compounds in practical applications and provide suggestions for future research directions. Additionally, we will discuss the possibility of enhancing antioxidant properties through the synthesis of derivatives, offering a valuable perspective for future studies.

Round 2
Reviewer 2 Report
Comments and Suggestions for Authors
The paper was revised carefully and could be accepted.